# Notched Behaviors of Carbon Fiber-Reinforced Epoxy Matrix Composite Laminates: Predictions and Experiments

**Shaoyong Cao [1], Yan Zhu [2] and Yunpeng Jiang [2,*]**

1   School of Industrial Automation, Beijing Institute of Technology, Zhuhai 519088, China; lemonsayong2005@163.com
2   College of Aerospace Engineering, Nanjing University of Aeronautics and Astronautics, Nanjing 210016, China
*   Correspondence: ypjiang@nuaa.edu.cn; Tel.: +86-25-84893240

**Abstract:** This paper experimentally studied the influence of the notch shape and size on the damage evolution and failure strength (tension and torsion) of carbon fiber-reinforced epoxy matrix (CFRP) laminates. Hashin's damage criteria were utilized to monitor the evolution of multi-damage modes, and FEM simulations were also performed by using the ABAQUS code to clarify the specific damage modes in detail as an instructive complement. The failure characteristics of all the notched samples were analyzed and compared with those without notches. The measured results presented that the existence of a variety of notches significantly impaired the load carrying capacity of CFRP laminates. The tensile strengths of C-notch and U-notch increase with an increasing notch radius, while the ultimate torques of C-notch and V-notch decrease with an increasing notch size and angle. The variation in notched properties was explained by different notch shapes and sizes, and the failure characteristics were also presented and compared among notched CFRP laminates with varied notches.

**Keywords:** carbon fiber-reinforced polymers (CFRPs); notch effect; finite element method (FEM); damage propagation; mechanical behaviors





## 1. Introduction

CFRP laminates have been widely applied in the aerospace, aviation and automobile industries due to their high specific stiffness, high specific strength and high design freedom. These composite laminate structures usually include discontinuities such as cut-outs for access and fastener holes for joining and inevitably become vulnerable regions under thermo-mechanical loading [1–3]. Understanding their notched behaviors is necessary for designing these complex structures, in which various parts are mostly connected with bolts and rivets [4]. The effect of these discontinuities on the mechanical performances is an important issue because it causes a relatively large reduction in strength compared to the unnotched laminates. Koricho et al. [5] proposed an innovative solution of using the 'tailored placement' of fibers around the holes/notch to make fabric laminates without the need for the drilling and machining of holes, thereby eliminating the sources of delamination. After drilling a hole, the tensile strength sharply decreased from 1012 MPa to 385 MPa, while the strength by adopting the developed processing could maintain 86% of the intact specimens.

A great number of experiments have been performed in studying the failure strength of notched CFRP laminates containing various shapes and sizes of notches under tensile, compressive and multi-axial loadings. Belgacem et al. [6] tested the center-notched CFRP laminates with radii of 2 mm, 6 mm and 10 mm and found that the ultimate strength is reduced by the order of 23%, 26% and 45% compared with those without notches. Torabi et al. [7] investigated the load-carrying capacity of glass/epoxy laminates with central U-shaped notches of various tip radii by using the virtual isotropic material concept

without the need for ply-by-ply failure analysis. Xu et al. [8] studied the size effect in center-notched CFRP laminates under compression and found that the compressive strength of the small center-notched specimen is similar to that of the open-hole specimens. As the in-plane sizes increase, the center notches are weaker than the open holes. Serra et al. [9] carried out numerical and experimental studies on the size effect in notched CFRP laminates using the discrete ply modeling method [10] and showed that their strength reduces with an increasing specimen size. Lee et al. [11] considered the notch size, ply and laminate thickness to be the most important variables of scaling effects on the strength and found that the strength reduction is due to the hole size effect rather than the specimen thickness or volume increase. Wan et al. [12] studied the notch effect on the strength and fatigue life of double edge notched laminates and declared that the notch effect is strongly dependent on the fiber type, notch depth, load type and load sequence. Torabi et al. [13] also studied the E-glass/epoxy composites weakened by blunt V-notches with different notch angles and tip radii. Ghezzo et al. [14] analyzed the interaction between two holes set in different configurations with respect to the load direction to seek the minimum distance at which there is no superposition of notch effects within the area between two near holes.

Llobet et al. [15] measured the strength of CFRP notched laminates under static, tension–tension fatigue and residual strength tests. Lagattu et al. [16] characterized over-stress accommodation which develops near the notches. Vieille et al. [17] studied the influence of the matrix nature on the tensile thermo-mechanical behavior of notched laminates and indicated that the hole is an open access through the thickness for the heat flux causing thermal degradation and decreasing the laminate tensile properties. The tensile strength decreases from 472 MPa to 247 MPa after introducing a central hole, as for C/PPS laminates. Ye et al. [18] measured the residual strength of notched cross-ply CFRP laminates with different fiber/matrix adhesion and indicated that laminates with poor interfacial adhesion exhibit a higher residual strength than those with strong adhesion. Furthermore, the effect of interfacial adhesion on the fatigue residual strength of circular notched laminate was studied [19]; the strengths reduce from 1000 MPa to only 300 MPa with an increasing hole diameter. Czél et al. [20] prepared CFRP laminates with hybrid modulus carbon fibers and showed that reduced notch sensitivity was demonstrated for open holes and sharp notches. Qiao et al. [21] studied the failure behavior of notched laminates under multiaxial quasi-static and fatigue loading, and a significant non-linearity in the stress-strain curves was exhibited, becoming more and more significant with increasing shear load components.

The progressive damage analyses are usually employed on the notched behaviors of CFRP laminates. Liu [22,23] introduced the nonlocal integral theory into the damage model to solve the localization problem of composites and derived an FEM model on the nonlocal intra-laminar damage and interlaminar delamination of laminates. Hu et al. [24] studied the layer-by-layer stress components and the damage propagation and failure in notched laminates by the theoretical method and FEM. Divse et al. [25] investigated the stress concentrations factor, damage progression and tensile notched strength of CFRP laminates and presented that the FEM plane model slightly underestimated the extent of damage propagation and failure load when compared with a 3D progressive damage model. Ng et al. [26,27] proposed a progressive failure analysis together with a newly developed maximum notched strength method and confirmed that the location of failure initiation for laminates with large hole sizes is different from that for laminates with smaller holes. Riccio et al. [28] presented that the ABAQUS plane stress gradual degradation model overestimates the damage accumulation leading to a premature ultimate load, and the 3D degradation model could correctly predict the mechanical response and ultimate load. Maa et al. [29] combined the generalized standard material model with the principal damage concept of composite materials. Laurin et al. [30] presented a simplified strength analysis method for perforated plates with open-holes, ensuring design office requirements in terms of precision and computational time. Chen et al. [31] predicted the tensile and compressive strengths of notched laminates by extending Whitney and Nuismer's average

stress failure criterion. Morgan et al. [4] developed a more complete picture of notch effects and analyzed the accuracy of the theory of critical distances.

In the light of the above literature survey, the purpose of this work is to reveal the effect of the notch size and shape on the mechanical behaviors of CFRP laminates under tension and torsion; especially, the torsion testing on the notched CFRP seems very limited at the present time and insufficient for structural engineering. A joint method of an experiment and FEM is used to understand the damage mechanism in these notched samples. The variation in tension and torsion properties is tested by using CFRP laminates containing varied notch shapes and sizes.

In Section 2, we present the preparation of samples for tensile and torsion testing. The user subroutine and FEM modeling are then presented in Section 3. The results of both measured and simulated failure mechanisms under tension are discussed in Section 4. The results of both tested and simulated failure mechanisms under torsion are discussed in Section 5. The conclusion is finally given in Section 5.

## 2. Materials and Methods

### 2.1. Materials

T700 carbon plain fabric (areal weight: 300 g/m$^2$, and thickness: 0.125 mm) is supplied from Japan Toray Industries, Inc. The epoxy resin used is E51 (Nanya epoxy Co., Ltd., Kunshan, China) with a low viscosity of 600–800 MPa·s at room temperature, and Polyamide 650 is used as the curing agent with a ratio of 30:10 parts in weight. The technical data of the Resin are listed in Table 1.

**Table 1.** Specification of epoxy resin.

| Category | Parameters |
|---|---|
| Product categories | Bisphenol A epoxy vinyl ester resin |
| Appearance | Colorless |
| Viscosity | 600–800 MPa·s, 25 °C |
| Tensile strength | 85 MPa |
| Hardness (ShoreD) | 88 |
| Density | 1.05 g/cm$^3$ |
| Compress strength | 300 MPa |
| Elastic modulus | 1.0 GPa |
| Poisson's ratio | 0.38 |

### 2.2. Manufacturing of Composites

The epoxy system is synthesized by mixing the resin with a hardener at the weight ratio of 30:10. The cross-ply [0/90/0/90]$_s$ laminates are stacked by hand lay-up, fabricated using the vacuum resin infusion process (ECVP425 vacuum pump, Easy Composites Asia Ltd., Beijing, China) and finally cured at room temperature for the duration of 24 h. Figure 1 illustrates the production process for CFRP laminate panels. The resulting volume fraction of the carbon fiber is 60% for all composites. The nominal thickness of the samples is measured as 1.0 ± 0.1 mm, The diamond tip water-cooled saw blade (ONEJET50-G30 × 15 Waterjet cutting machine, OneJet Co., Ltd., Shenzhen, China) is used to cut these CFRP panels into flat coupons with the specific dimensions and shapes of the testing samples shown in Figure 2, which also illustrates the photographs of various test samples. In order to study the notched behaviors of CFRP laminates, tension and torsion tests are conducted using specimens that are cut from the [0/90/0/90]$_s$ panels using a water jet cutter.

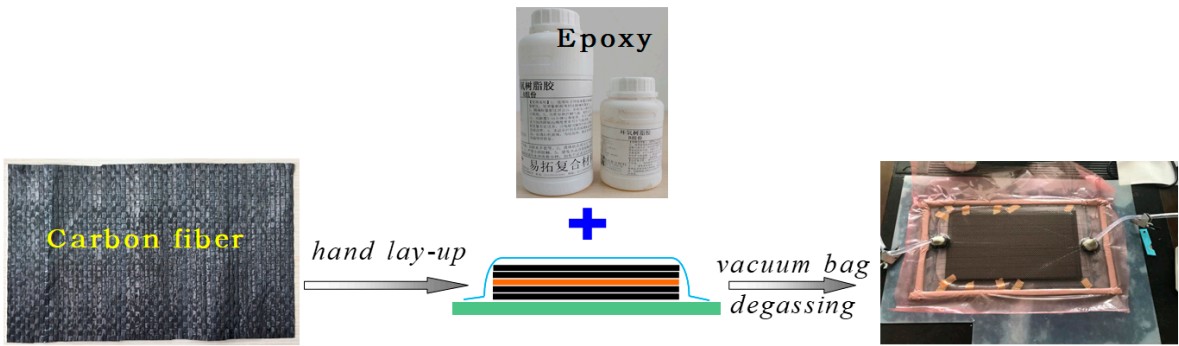

**Figure 1.** Production processing of composite samples.

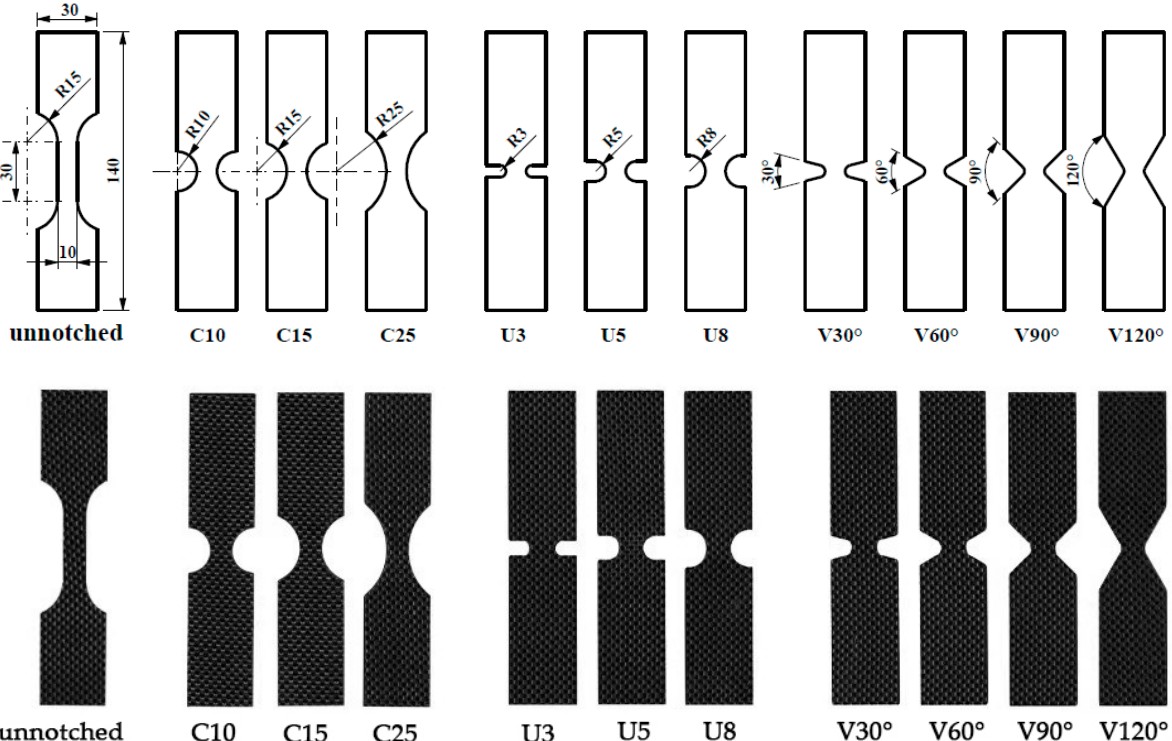

**Figure 2.** Specimen design of unnotched and various notched specimens, and photographs before testing.

### 2.3. Tension Tests

Tension tests are performed according to ASTM D638 for testing tensile properties of plastics, using a hydraulic-driven Instron testing machine 3365 series (Boston, MA, USA) and data acquisition card AD12 (Contec Corporation Ltd., Osaka, Japan). Tension tests are carried out by using a hydraulic-driven Instron tension tester at a constant crosshead displacement speed of 0.5 mm/min at room temperature, according to ASTM D3039 used for testing the tensile properties of polymer matrix composite materials. An extensometer with a gauge length of 50 mm is attached to the specimen to measure the average longitudinal strain. Three specimens of each composition are tested, and the average value is reported.

### 2.4. Torsion Tests

The specimens for the torsion test are similar to those of the tension test and measured using the electronic torsion testing machine NJ-S series (Beijing Timesun measurement and control technology Co., Ltd., Beijing, China). The cross-head speed for torsion tests is 10°/min, and Figure 3 shows a schematic diagram for the torsion test frame and the associated sample position in the frame.

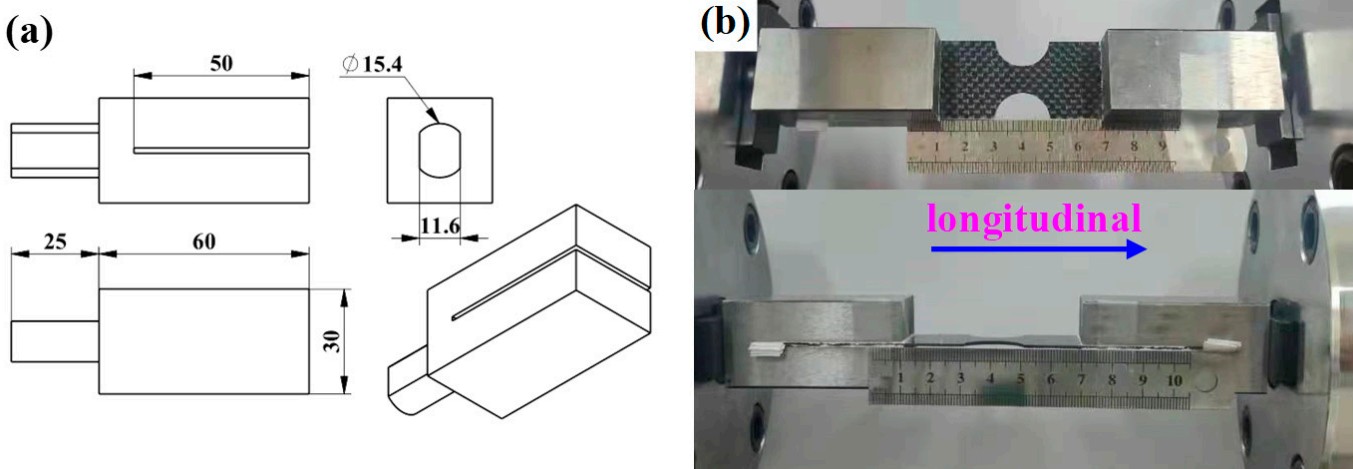

**Figure 3.** Schematic illustration of the torsion test frame in (**a**) and the sample position in the frame in (**b**); here, the upper picture is the top view, and the lower picture is the front view.

## 3. FEM Simulation

Eight-node quadrilateral, linear, thick-shell elements with six degrees of freedom per node are used. The user material subroutine (UMAT) is incorporated into the ABAQUS [32], and the geometric non-linearity is considered in the damage analysis. The geometric non-linearity and large deformation are accomplished by using the incremental loading and the NLGEOM parameter on the "STEP" option in ABAQUS. Damage modes in laminate structures strongly rely on the ply orientation, loading direction and panel geometry. There are four basic modes of failure that occur in a laminate structure. These failure modes are: matrix cracking, fiber–matrix shear failure, fiber failure and delamination. De-lamination failure is not considered here due to the high complication in modeling the interaction between plies. Mesh convergence is tested to ensure computational accuracy. To simulate the failure modes (matrix failure in tension or compression, fiber–matrix shear failure and fiber crack), the elastic properties are made linearly dependent on four field variables. The first field variable represents the matrix failure index, the second represents the fiber–matrix shear failure index and the third represents the fiber crack. All the elastic coefficients are dependent on the field variables to reflect the development of damage in the composites.

The finite element implementation of this progressive failure analysis is developed for the ABAQUS structural analysis program using the user subroutine USDFLD. ABAQUS calls the USDFLD subroutine at all material points of elements that have material properties defined in terms of the field variables. The subroutine provides access points to a number of variables, such as stresses, strains, material orientation, current load step and material name, all of which can be used to compute the field variables. Stresses and strains are computed at each incremental load step and evaluated by the failure criteria to determine the occurrence of failure and the mode of failure.

In this work, the subscript '−1' stands for the fiber direction, and subscript '2' is the direction perpendicular to the fiber direction. The Hashin's damage criteria [33] are adopted in order to simulate damage growth in each ply, and the failure criteria are written as

$$\text{Fiber failure in tension } (\sigma_1 \geq 0): \ e_f^2 = \left(\frac{\sigma_1}{S_{1T}}\right)^2 + \frac{2\tau_{12}^2/G_{12}^0 + 3\alpha\tau_{12}^4}{2S_{12}^2/G_{12}^0 + 3\alpha S_{12}^4} \tag{1}$$

$$\text{Fiber failure in compression } (\sigma_1 < 0): \ e_m^2 = \left(\frac{\sigma_1}{S_{1C}}\right)^2 + \frac{2\tau_{12}^2/G_{12}^0 + 3\alpha\tau_{12}^4}{2S_{12}^2/G_{12}^0 + 3\alpha S_{12}^4} \tag{2}$$

$$\text{Matrix failure in tension } (\sigma_2 \geq 0): e_m^2 = \left(\frac{\sigma_2}{S_{2T}}\right)^2 + \frac{2\tau_{12}^2/G_{12}^0 + 3\alpha\tau_{12}^4}{2S_{12}^2/G_{12}^0 + 3\alpha S_{12}^4} \tag{3}$$

$$\text{Matrix failure in compression } (\sigma_2 < 0): e_m^2 = \left(\frac{\sigma_2}{S_{2C}}\right)^2 + \frac{2\tau_{12}^2/G_{12}^0 + 3\alpha\tau_{12}^4}{2S_{12}^2/G_{12}^0 + 3\alpha S_{12}^4} \quad (4)$$

$$\text{Fiber/matrix shear failure}: e_{fm}^2 = \left(\frac{\sigma_1}{S_{1C}}\right)^2 + \frac{2\tau_{12}^2/G_{12}^0 + 3\alpha\tau_{12}^4}{2S_{12}^2/G_{12}^0 + 3\alpha S_{12}^4} \quad (5)$$

where the factor $\alpha = 0.8 \times 10^{-14}$; these equations are used to monitor the evolutions of damage modes in each lamina, and the adhesion between laminas is assumed to be perfect, with no interface separation. The values of the field variables are set to be equal to zero in the undamaged state. After the failure index has exceeded 1.0, the associated user-defined field variable is set to be equal to 1. The mechanical properties in the damaged area are reduced appropriately, according to the property degradation model. The corresponding modulus and Poisson's ratio are assigned with a minimal value after the damage happens during the deformation. These damage modes are characterized by some internal state variables (SDV) as the output parameters.

To simulate the above failure modes, the elastic properties are made to be dependent on three field variables, *FV1*, *FV2* and *FV3*. Four variables represent the matrix failure, fiber/matrix shearing failure and fiber failure, respectively. The values of the field variables are set to be equal to zero in the undamaged state. After a failure index has exceeded 1.0, the associated user-defined field variable is set to be equal to 1.0. The mechanical properties in the damaged area are reduced appropriately, according to the property degradation model defined in Table 2. For example, if the matrix failure criterion is satisfied, namely, $\sigma_2 \geq 0$, $e_m \geq 1$, then only $E_{22} \rightarrow 0$ and $\nu_{12} \rightarrow 0$ are degraded, while $E_{11}$, $G_{12}$, $G_{13}$ and $G_{23}$ equal the initial value. It is noted that after a damage happens, the iteration increment will be automatically selected and decreased gradually while increasing the damage degree. As the iteration increment is less than the predetermined minimum increment size, the computation process will be terminated correspondingly.

**Table 2.** The stiffness degradation rules based on the failure state (1—fiber direction, 2—transverse direction, FV1—matrix failure, FV2—fiber/matrix shear failure, FV3—fiber failure).

| Intact | Matrix Failure | Fiber–Matrix Shear | Fiber Failure | All Failure Mode |
|---|---|---|---|---|
| $E_{11}$ | $E_{11}$ | $E_{11}$ | $E_{11} \rightarrow 0$ | $E_{11} \rightarrow 0$ |
| $E_{22}$ | $E_{22} \rightarrow 0$ | $E_{22}$ | $E_{22}$ | $E_{22} \rightarrow 0$ |
| $\nu_{12}$ | $\nu_{12} \rightarrow 0$ | $\nu_{12} \rightarrow 0$ | $\nu_{12} \rightarrow 0$ | $\nu_{12} \rightarrow 0$ |
| $G_{12}$ | $G_{12}$ | $G_{12} \rightarrow 0$ | $G_{12} \rightarrow 0$ | $G_{12} \rightarrow 0$ |
| $G_{13}$ | $G_{13}$ | $G_{13} \rightarrow 0$ | $G_{13}$ | $G_{13} \rightarrow 0$ |
| $G_{23}$ | $G_{23}$ | $G_{23}$ | $G_{23} \rightarrow 0$ | $G_{23} \rightarrow 0$ |
| $FV1 = 0$ | $FV1 = 1$ | $FV1 = 0$ | $FV1 = 0$ | $FV1 = 1$ |
| $FV2 = 0$ | $FV2 = 0$ | $FV2 = 1$ | $FV2 = 0$ | $FV2 = 1$ |
| $FV3 = 0$ | $FV3 = 0$ | $FV3 = 0$ | $FV3 = 1$ | $FV3 = 1$ |

## 4. Results of Tension and Discussion

### 4.1. Tension Testing Results

Figure 4 shows the stress–strain curves of unnotched samples, where the used material properties in the present simulations are quoted from the other work [34], which are given as: $E_1 = 140$ GPa, $E_2 = 10$ GPa, $\nu_{12} = 0.3$, $G_{12} = G_{13} = 4.0$ GPa, $G_{23} = 4.32$ GPa, $S_{1T} = 2180$ MPa, $S_{2T} = 87.1$ MPa and $S_{12} = 165$ MPa. The stiffness degradation method allows us to obtain the stress softening stage after the failure point in Figure 4. Moreover, these above material properties are verified by comparing the present numerical simulations with the measured stress–strain relations.

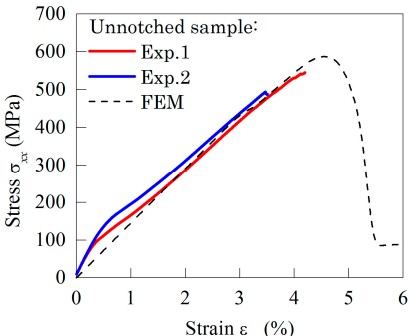

**Figure 4.** Stress–strain curves of unnotched samples under tensile loading (Exp.1 and Exp.2 denote two samples with the same sizes).

The mechanical characterization of the test samples in terms of tensile properties is conducted for different shapes and sizes of notches. The stress–strain relations are displayed in Figure 5, and it is noted that all samples show the brittle behaviors, exhibiting variations in both modulus and strength among these notched samples. Both numerical simulations and testing results present that all the circular notched samples exhibit similar stress–strain relations, and the C25 sample possesses the highest tensile strength, which is nearly similar to that of the unnotched samples. An increasing trend is found for tensile strength, suggesting that composite laminate is unaffected while increasing the notch diameter in such type of notched samples. The present predictions are in agreement with the measured results for the C-notched and U-notched samples, while the numerical calculations overestimate the tensile stiffness for the V-notched laminates and present lower predictions to their failure strengths. Furthermore, the stress–strain relations in the beginning stage of stretching are nonlinear, accompanying the marked transition from high stiffness to low stiffness. Such a change may originate from the inaccuracy in the ply orientation, off-centering clamp and sample cutting. The prediction errors in the ultimate strength between the numerical and measured results are in the range of 1~5% for the C-notch and range from 2% to 29.4% for the U-notch, while there is almost 14% underestimation for the V-notch.

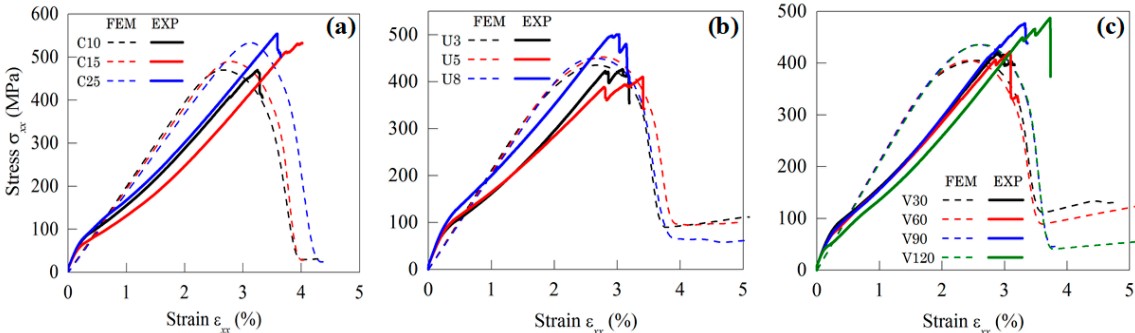

**Figure 5.** Stress–strain relations of C-notched samples in (**a**), U-notched samples in (**b**) and V-notched samples in (**c**).

Figure 6 shows the failure strength of C-notched, U-notched and V-notched samples; here, the errors are determined by measuring at least three specimens with the same size of notches. An increasing trend is observed for the tensile strength of the C-notch and U-notch specimens while increasing the notch radius, indicating that laminate structures become less sensitive while increasing the notch size. For the U-notched samples, it is noted that the changing tendency of strength with the notch radius is in accordance with what is found in Morgan's work [4]. In Morgan's research, the root radius of U-notched specimens changes from 0.5 mm to 20 mm, and the corresponding failure force increases from 4.19 kN up to 4.82 kN. Therefore, the dependence of the tensile strength on the root radius is the same

as in our results. On the other hand, the tensile strength changes slightly with the notch angle for the V-notched samples, implying that the V-notched laminate structures have high tolerance to the V-notched shape. Based on the comparisons between the numerical results and testing datum, the present simulations could predict the tested failure strengths for the C-notch and U-notch samples, while the adopted numerical method for the V-notch specimens under-estimate the measured results.

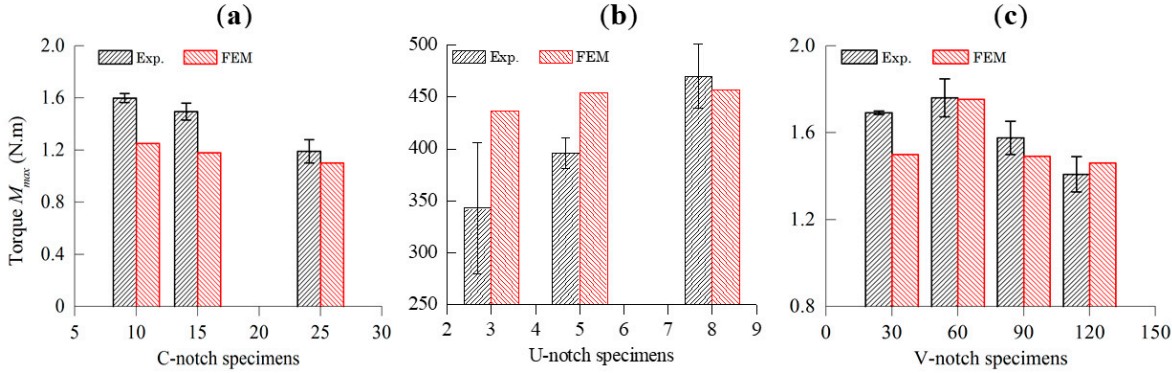

**Figure 6.** Dependence of the failure strength of the C-notched samples in (**a**), U-notched samples in (**b**) and V-notched samples in (**c**) with varied shapes.

### 4.2. Modeling of Tension Failure

Figure 7 shows the predicted damage modes in the C-notched, U-notched and V-notched specimens with varied sizes and angles. Here, fiber crack and shear damage are the two main failure modes in the laminate structures under tensile loading. It is observed that both the fiber crack and shear damage areas gradually become more and more wide along the C-shape notch while increasing the notch radius from 10 mm to 25 mm, and the severity of damage is relatively reduced; therefore, their corresponding tensile strengths increase with the decrease in these damage modes.

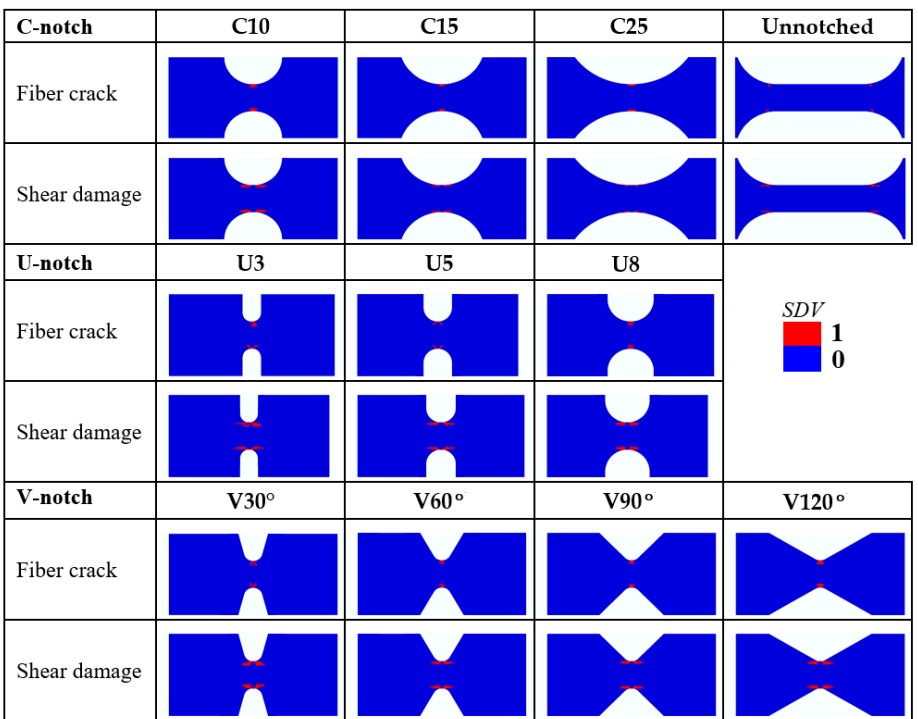

**Figure 7.** Damage modes in unnotched, C-notched, U-notched and V-notched laminates.

After examining the damage modes in the U-notched laminates in Figure 7, it is also found that both the fiber crack and shear damage areas are gradually reduced while increasing the notch radius from 3 mm to 8 mm, accompanied by an increment in their tensile strengths; therefore, the failure strength of the U-notched samples becomes closer to that of the unnotched sample.

Based on the damage contours of the V-notched samples in Figure 7, both the fiber crack and shear damage areas do not have an evident change in the samples with different notch angles from 30° to 120°, indicating that these notched laminates are insensitive to the notch angles, and, thus, their tensile strengths nearly remain unchanged.

### 4.3. The Damage Mechanism under Tension

The fractured specimens after tensile failure are demonstrated in Figure 8, and these failed samples display a brittle failure mode, which is attributed to the brittle nature of carbon fiber and epoxy resin. Long splitting, multi-mode fiber pull-out and delamination are observed. For the U8 and V120 samples, shear damage modes appear remarkably, and some carbon fibers along the transverse direction pull out, contributing to the higher loading capacity. A further increase in the notch radius results in the detriment of the laminates in terms of tensile strength. It is observed from failure modes that delamination formed in some local regions. These behaviors could be attributed to the brittle nature of the epoxy resin and the imperfect interphase cohesion between the carbon fiber and epoxy matrix. Failure mechanisms over the fracture surface indicate that some carbon fibers pull out, exhibiting the weak interfacial interaction induced by the vacuum infusion, and some micro-voids are still in existence. Moreover, the adhesion between the fiber and epoxy mainly stems from the mechanical friction between them, without any chemical bonding.

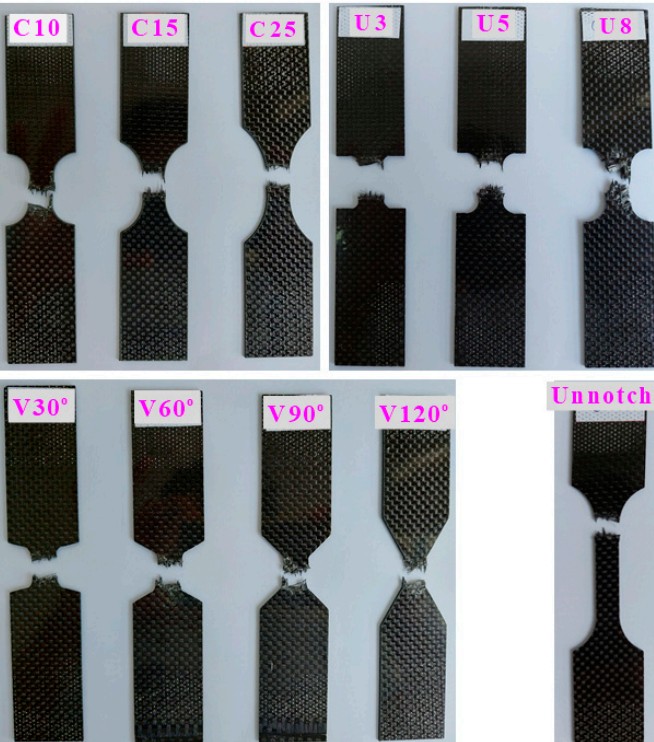

**Figure 8.** Failure surfaces of tension test samples.

## 5. Results of Torsion and Discussion

### 5.1. The Damage Mechanism under Tension

The mechanical characterization of torsion samples in terms of torsion properties is conducted for different shapes and sizes of notches. The torque–rotation angle relations are displayed in Figure 9; it is noted that all samples show the nonlinear behaviors, exhibiting

progressive damage evolution among these notched samples. All the circular notched samples show the same stress–strain relations; the tensile strength of the C25 sample exhibited the highest value, which is nearly similar to that of unnotched samples. The present predictions are in good agreement with the measured results for all the notched laminates. Especially, the staged degradation associating the progressive multi-damage modes in the laminates is well captured in the predicted torque–rotation angle curves. These predicted torque–rotation angle curves present marked oscillations in the torque at the failure stage, which lie in the damage accumulations during the torsion deformation.

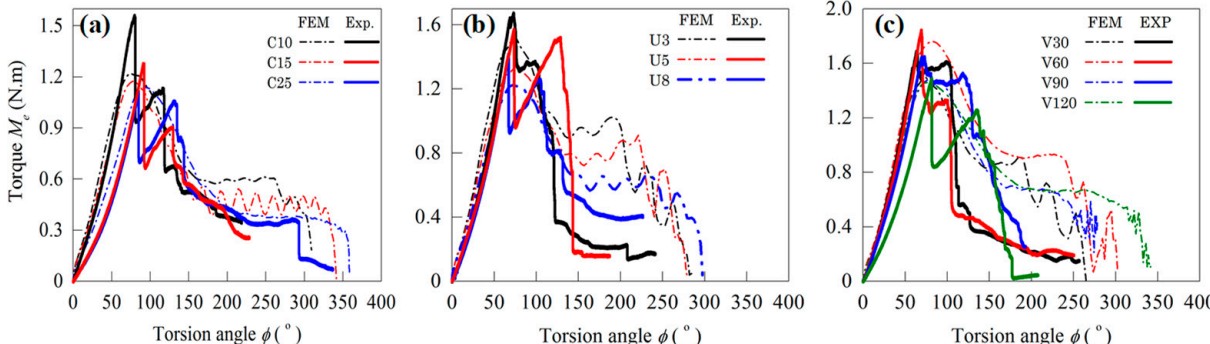

**Figure 9.** Torque–rotation angle (Me-φ) relations of the C-notched samples in (**a**), U-notched samples in (**b**) and V-notched samples in (**c**).

All the failure torques of these notched samples are summarized in Figure 10. The maximum torques of the C-notch and V-notch samples decrease with the increase in the notch size and angle, and the maximum torque becomes insensitive to the notch radius of the U-notched samples. The maximum torque decreases from 1.6 N·m to 1.2 N·m while increasing the notch radius in the C-notch specimens. Moreover, the tendency of the U-notched samples exhibits the 'U-shaped' variation, and the U5 sample processes the lowest torque among these specimens. Moreover, it should be mentioned that the changing tendency in the failure torque of the U-notched samples with the notch radius is also in agreement with that in Morgan's work [4]. In Morgan's research, the root radius of the U-notched specimens changes from 0.5 mm to 20 mm, and the corresponding failure torque increases from 2.76 N·m up to 3.66 N·m. Therefore, the dependence of the torsion strength on the root radius is the same as in our results.

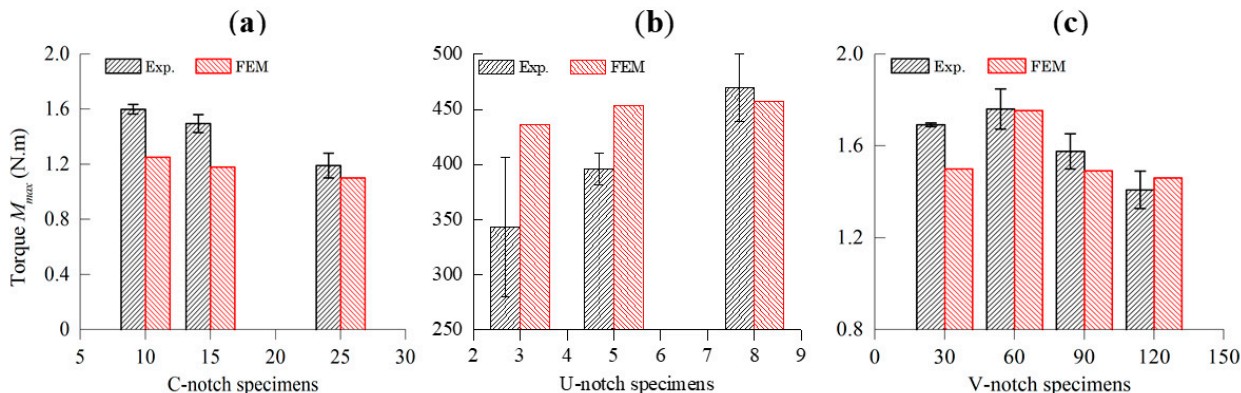

**Figure 10.** The measured and predicted failure torques of the C-notched samples in (**a**), U-notched samples in (**b**) and V-notched samples in (**c**).

Based on Hashin's damage criterion, the in-plane multi-damage modes in CFRP laminates can be well assessed by directly programing a user-subroutine in the ABAQUS code. However, the out-of-plane damage mode cannot be considered by this model, and thus, an additional cohesive zone with traction–separation laws should be involved to

consider delamination growth, especially for the complicated multiple failure progress in the torsion loading.

### 5.2. Modeling of Torsion Failure

Figure 11 shows the damage modes in C-notched, U-notched and V-notched specimens under torsion. The present samples under torsion loading deform in the form of a very complicated shape and usually experience a large rotation angle before the final failure owing to their low torsion stiffness $G \times I_p$ ($G$ is the shear modulus, and $I_p = \oint_A \rho^2 dA$ is the polar moment of inertia over the minimum transaction of the laminates, in which $\rho$ is the distance from the shaft center to the element of the infinitesimal area $dA$). The highly twisted deformation constitutes the main part of the torsion damage evolution, in which shear damage and matrix failure happen during the torsion deformation, and some difference in the notch zones among these samples could be observed. For instance, there is nearly no discrepancy in the damage zones for the V-notched samples, and correspondingly, their maximum torques are also similar.

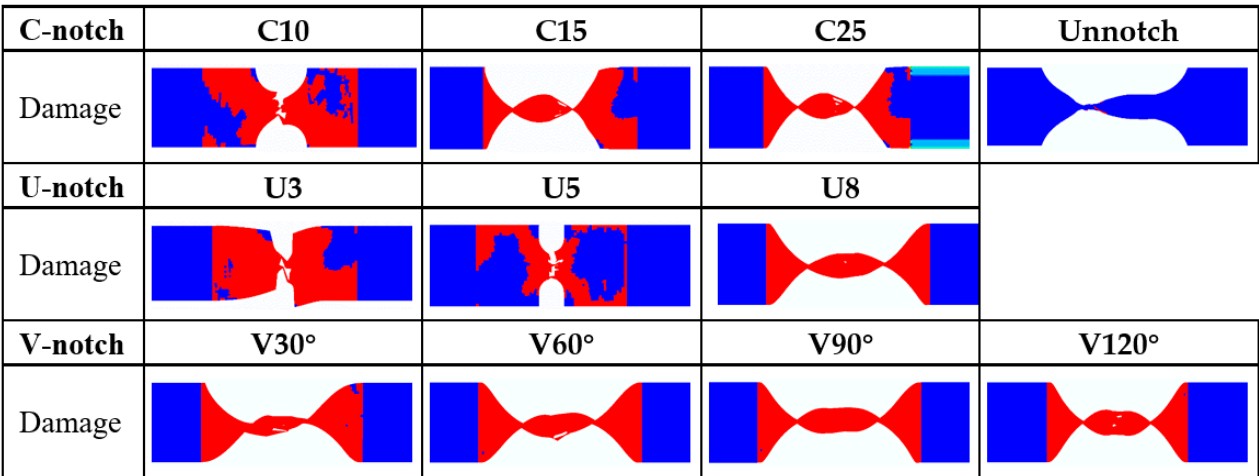

**Figure 11.** Damage mode by FEM simulations; red designates *SDV* = 1, and blue designates *SDV* = 0 in these contours.

### 5.3. The Torsion Mechanism

Figure 12 shows the failure surfaces of the torsion test samples, and it is observed that all the samples that fail in the form of fiber cracking follow a 45° spiral plane. The failure could be explained by the classic stress analysis, and the ±45° section plane is the principal plane in which the first principal stress $\sigma_1$ and third principal stress $\sigma_3$ lie. $\sigma_1$ would increase rapidly up to the critical strength of fibers under the torsion loading and then finally break the longitudinal fibers along this direction. The shear stress along the layers of the laminate structures under torsion also leads to the delamination damage between the plies due to their limited inter-laminar adhesive strength, which could be found from these failure surfaces in Figure 12. These behaviors could be attributed to the brittle nature of the sample, indicating imperfect interphase adhesion between fiber–epoxy interactions. Additionally, a striking permanent unrecoverable plastic deformation in the out-of-plane direction after the torsion loading is observed in all the samples, stemming from the progressive damage in the plies with the increase in the rotation angle.

Torsion provides a more complete picture of composite deformation for specific service scenarios as a complex deformation mode. However, the analysis of multi-damage progression under torsion should be challenging for CFRP laminates, which demands a comprehensive understanding in the next work.

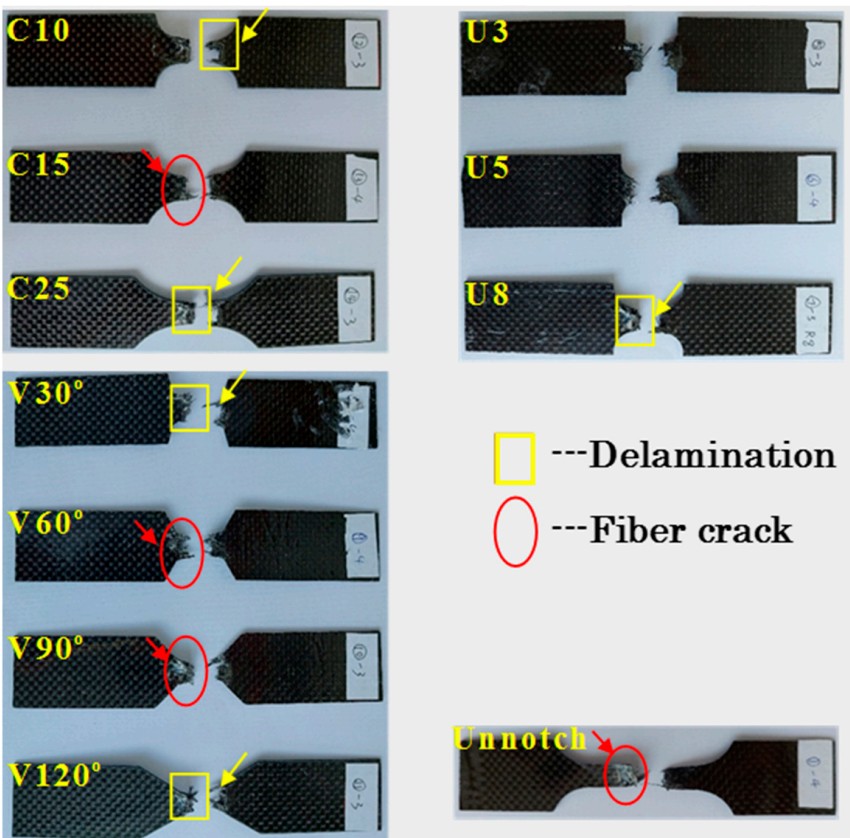

**Figure 12.** Failure surfaces of torsion test samples.

## 6. Conclusions

In this study, the mechanical characterization of the notched carbon fiber/epoxy composite laminates is investigated by incorporating notches with different shapes and sizes. The results indicate that the incorporation of notches significantly decreased the mechanical properties of carbon fiber/epoxy composites due to the fact that the stress concentration effect as a result of local stress changes with different sizes and shapes of notches. Several main conclusions are reached:

1. The tensile strengths of the C-notch and U-notch specimens increase while increasing the notch radius, and the tensile strength changes slightly with the notch angle for the V-notch samples.
2. The maximum torques of the C-notch and V-notch samples decrease while increasing the notch size and angle, and the maximum torque becomes insensitive to the notch radius of the U-notched samples.
3. All the notched samples displayed brittle failure modes, long splitting, multimode fiber pull-out and delamination. Shear damage modes appear remarkably, and some carbon fibers along the transverse direction pull out, contributing to the higher loading capacity.

**Author Contributions:** Conceptualization, Y.J. and Y.Z.; methodology, Y.Z.; validation, S.C.; formal analysis, S.C. and Y.Z.; writing—original draft preparation, Y.J.; writing—review and editing, S.C.; supervision, Y.J.; funding acquisition, Y.J. All authors have read and agreed to the published version of the manuscript.

**Funding:** This research was funded by the Fundamental Research Funds for the Central Universities, grant number NS2022012.

**Data Availability Statement:** The data that support the findings of this study are available from YPJ upon reasonable request.

**Conflicts of Interest:** The authors declare no conflict of interest.

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
