# Peer review of "Notched Behaviors of Carbon Fiber-Reinforced Epoxy Matrix Composite Laminates: Predictions and Experiments"

_jcs, doi:10.3390/jcs7060223_

Round 1

Reviewer 1 Report

In this work, the authors made an investigation on the mechanical behaviors of notched carbon fiber composite laminate specimens under tension and torsion. The experimental findings were some useful addition to the composite community, whereas the numerical simulations were lack of sufficiency. Significant revision is required before a publication recommendation can be made. Specific comments are as follows.

1.     FEM based failure analysis for a laminated composite with notch in this paper involved many issues that were not clear. First, whether Eqs. (1)-(5) were applied to a lamina (each layer of the laminate) or the whole laminate? If they were applied to a lamina, a stiffness discount scheme should be adopted and an ultimate failure condition for an element of the laminate should be incorporated. None of them was shown. If they were applied to the laminate, which must have been assumed by the authors as a homogeneous anisotropic material, the mechanical property parameters given in between the lines 184 and 185 were not pertinent, since the laminate was in-plane isotropic (E2=E1). Second, what was the ultimate failure condition applied to the laminate with a notch, which should be considered as a composite structure? In other words, which element’s failure corresponded to the ultimate failure of the laminate under which the ultimate strength of the laminate was predicted? As demonstrated in a recent publication in Composite Structures (306: 116558, 2023), the ultimate failure of an element within the neighborhoods of singularity and weak singularity points should be excluded, since the stresses calculated at those points are untrue. Third, generally a stress-strain curve of the composite up to the ultimate failure, i.e., at the highest stress point, is able to be predicted and automatically terminated in an incremental solution process. The authors’ predictions in this work, however, were lasted quite after the highest stress point. Then, what was the termination condition used by the authors? If no experimental measurement had been known in advance, could any of the authors’ predictions be carried out? Fourth, a useful prediction must be able to be made independently. Namely, the input data for the prediction must be able to be prepared in advance. It seemed that the input data in the lines of 184 and 185 were for a unidirectional lamina. Why did not the authors measure them from the unidirectional lamina [0] specimens? Furthermore, the E2 data was apparently much higher than the transverse modulus of the unidirectional lamina.   

2.     The modulus and Poisson’s ratio of the epoxy resin need to be added into Table 1.

3.     The definitions for the material parameters in the lines of 184 and 185, especially S1T, S2T, and S12 should be given. If S2T and S12 stand for the transverse tensile and in-plane shear strengths of a unidirectional lamina, the data given were apparently not true.

Author Response

Correspondence to the reviewers’ comments

  1. FEM based failure analysis for a laminated composite with notch in this paper involved many issues that were not clear. First, whether Eqs. (1)-(5) were applied to a lamina (each layer of the laminate) or the whole laminate? If they were applied to a lamina, a stiffness discount scheme should be adopted and an ultimate failure condition for an element of the laminate should be incorporated. None of them was shown. If they were applied to the laminate, which must have been assumed by the authors as a homogeneous anisotropic material, the mechanical property parameters given in between the lines 184 and 185 were not pertinent, since the laminate was in-plane isotropic (E2=E1). Second, what was the ultimate failure condition applied to the laminate with a notch, which should be considered as a composite structure? In other words, which element’s failure corresponded to the ultimate failure of the laminate under which the ultimate strength of the laminate was predicted? As demonstrated in a recent publication in Composite Structures (306: 116558, 2023), the ultimate failure of an element within the neighborhoods of singularity and weak singularity points should be excluded, since the stresses calculated at those points are untrue. Third, generally a stress-strain curve of the composite up to the ultimate failure, i.e., at the highest stress point, is able to be predicted and automatically terminated in an incremental solution process. The authors’ predictions in this work, however, were lasted quite after the highest stress point. Then, what was the termination condition used by the authors? If no experimental measurement had been known in advance, could any of the authors’ predictions be carried out? Fourth, a useful prediction must be able to be made independently. Namely, the input data for the prediction must be able to be prepared in advance. It seemed that the input data in the lines of 184 and 185 were for a unidirectional lamina. Why did not the authors measure them from the unidirectional lamina [0] specimens? Furthermore, the E2 data was apparently much higher than the transverse modulus of the unidirectional lamina. 

Answer:

(1) These equations are used to monitor the evolutions of damage modes in each lamina, and the adhesion between laminas is assumed to be perfect with no interface separation. The corresponding modulus and Poisson’s ratio are assigned with a minimal value after the damage happens during the deformation. These damage modes are characterized by some internal state variables (SDV) as the output parameters.

(2) The damage mechanics was used in the present work, and stiffness degradation method was adopted to describe the failure progress in CFRPs, whereby the ultimate strength of the whole laminate could be achieved.

(3) Since the damage mechanics belongs to continuum medium method, the continuous stress-strain curves could be calculated by the present model. The stiffness degradation strategy affects the development of stress-strain response after the damage happens.

(4) The used carbon fibers are T700, and epoxy resin is E51 type in the present work. The fundamental properties of these constituents could be readily found in many references, and thus we referred other’s testing datum to avoid the standard tests in measuring these mechanical coefficients.

2. The modulus and Poisson’s ratio of the epoxy resin need to be added into Table 1.

Answer:

Added

3. The definitions for the material parameters in the lines of 184 and 185, especially S1T, S2T, and S12 should be given. If S2T and S12 stand for the transverse tensile and inplane shear strengths of a unidirectional lamina, the data given were apparently not true.

Answer:

After checking the numerical results, and these errors were modified as:

E1=140GPa, E2=10GPa, v12=0.3, G12=G13=4.0GPa, G23=4.32GPa, S1T=2180MPa, S2T=87.1MPa and S12=165MPa.

Reviewer 2 Report

Dear Authors, 

the submitted manuscript "Notched Behaviours of Carbon Fibre Reinforced Epoxy Matrix Composite Laminates: Predictions and Experiments" addresses the effect of notch size and shape on the failure behavior of CFRP materials. 

The research theme is quite classic which means have been studied for many decades. Moreover, the tensile strength of FEM results deviates from experimental data so far. 

The result seems so simple and inevitable to me, thus this reviewer suggest reject.

Author Response

we carefully modified this paper

Reviewer 3 Report

This work utilizes a joint experimental and simulation method to investigate the tensile and torsional behaviors of notched carbon fiber/epoxy composite laminates with different shapes and sizes. Failure modes and other characteristics of the composite laminates are presented, analyzed, and compared among the notched samples. The results indicate that the notches can significantly reduce the load carrying capacity of the composite laminate structures, and the variations in mechanical behaviors and properties are attributed to different shapes and sizes. While the structure and writing of this manuscript are fine, it could be strengthened by a more explicit discussion of fracture mechanisms as well as its broader implications. On the other hand, there are many overly general, vague, or ambiguous terms and descriptions in the main sections, and the Method section reads incomplete and lacks important technical details. Moreover, in many cases the figures, data presentations, and equation presentations fall short of expectations, e.g., missing legends, labels, color codes for subfigures and captions, missing descriptions for symbols and parameters in equations, etc. I would recommend “Major Revision” for this manuscript, and the authors are expected to address the following issues and questions before the next submission.

1.     Page 1, Line 9 – 18

The Abstract does not provide sufficient details on the specific findings or implications of the study. I suggest that the authors should add a sentence or two to discuss the main results (with specific values reported) as well as the broader impact of this research to make the Abstract more informative.

2.     Page 1 – 2, Line 34 – 72

(1)  The authors presented several experimental studies examining the strength reduction of notched CFRP laminates. However, the authors did not report specific numerical values regarding the extent to which the strength was reduced.

(2)  The authors are expected to limit their literature review on the residual strength of notched CFRP laminates with respect to shape and size under different loadings to the most recent five years, in order to ensure that the references accurately reflect the current state of knowledge in the field.

3.     Page 3, Line 110 – 120

In the subsection titled “Manufacturing of composites”, the authors did cover the manufacturing process and testing method for producing CFRP laminates, but additional details regarding the materials, equipment, and specific testing procedures are expected to be included to enhance the understanding of such a process. For instance, elaborating on the type of equipment used for vacuum resin infusion process.

4.     Page 4, Line 124 – 127

Figure 2 and Figure 3 appear to be duplicated, since the shape of each notched specimen can be inferred from Figure 2. Maybe the authors can only keep Figure 2 or merge these two figures into one.

5.     Page 4, Line 129 – 135

Can the authors elaborate on the meanings of ASTM and D638 standards?

6.     Page 4, Line 142 – 144

(1)  Figure 4 should include labels for subfigures. For example, subfigure labels (a), (b), (c), and (d) could be added to show different views of the specimen. Axes indicating longitudinal and transverse directions could also be added.

(2)  Readers are unable to readily discern the type of test being conducted or distinguish between the top right and bottom right schematics. To address this, the authors may consider adding subfigure labels (e) and (f) (and the associated descriptions in the caption), as well as double-ended arrows indicating the direction of torsion in these subfigures.

7.     Page 5, Line 145 – 167

(1)  Can the authors provide visualizations of the mesh they used in their FEM models? Specifically, I am interested in how the authors treated discontinuities in the notched areas and cracks.

(2)  What are the strain rates and the torsional angle rates used in the FEM simulations? Are these parameters and setups comparable with those in the actual experiments?

(3)  Can the authors provide an explanation for why delamination failure was not taken into account?

8.     Page 5, Line 170 – 174

The authors are expected to provide detailed explanations for symbols and parameters that appear in equation (1) through (5).

9.     Page 6, Line 181 – 187

(1)  The authors wrote, “Comparing the predictions with the measured results could determine the basic material properties, …”. It is not clear how the authors compared the predictions with the measured results.

(2)  Are the material properties provided in Line 184 – 185 obtained through experimentation? If yes, then how did the authors calculate the Young's modulus given that the stress-strain curves appear to have a non-linear elastic region? Did the authors perform linear regression or take the second derivatives of the polynomial fitting to calculate it?

(3)  There is no explanation to tell the difference between Exp. 1 and Exp. 2 in Figure 5.

(4)  Can the authors provide an explanation as to why the experimental curves cease to exhibit further data prior to the point of failure in Figure 5?

10.  Page 6, Line 188 – 198

(1)  The authors stated, “…, tensile strength of C25 sample exhibited the highest value that is nearly similar to that of unnotched samples.” Is this conclusion drawn from experiments or from FEM simulations? Since the experimental ultimate strength values were not presented or discussed. The authors should report all these kinds of specific values in the discussions throughout Section 4.

(2)  The authors stated, “…, while the numerical calculations overestimate the tensile stiffness for the V-notched laminates, and present lower predictions to their failure strengths.” The authors should conduct a detailed examination and discussion on the possible factors that contribute to the overestimation of stiffness. Moreover, the authors should provide more information on the potential errors or uncertainties from both the experiments and the simulations.

11.  Page 6, Line 199 – 200

The labels are included in subfigures of Figure 6, but the letters do not appear in the caption.

12.  Page 6, Line 204 – 205

The authors mentioned that “the changing tendency of strength with the notch radius is in accordance with that is found in Morgan’s work”. Can the authors provide a clear explanation of the changing tendency in this context? Additionally, can the authors elaborate on the findings from Morgan's work? The current discussion section reads overly general and ambiguous, so a more informative and specific discussion would be expected.

13.  Page 7, Line 211 – 212

Page 9, Line 257 – 258

(1)  Can the authors modify the x-axis ticks for Figure 7 (b) and Figure 11 (b)? The authors omitted 3 and 5, so readers cannot identify U-3 and U-5 from the horizontal ticks in Figure 7 (b) and Figure 11 (b).

(2)  Again, the authors should include the letter labels in the captions of Figure 7 and 11.

(3)  The authors presented the bar plots with errors in Figure 7 and Figure 11, but there is no description provided regarding the methods/steps to calculate these errors, e.g., the number of samples considered in the statistical calculations.

14.  Page 7, Line 213 – 219

(1)  Figure 8 lacks a color code or legend to indicate the corresponding representation of the blue and red regions.

(2)  The authors stated, “Figure 8 shows the predicted damage modes in C-notched specimens”. However, Figure 8 also shows the predicted damage modes in U-notched and V-notched laminates. In addition, the authors should discuss how the positions and distributions of the damage modes were predicted.

(3)  The authors stated, “both fiber crack and shear damage areas gradually reduced with increasing the notch radius from 10 mm to 25 mm”. It is hard to see the fiber crack and shear damage areas gradually reducing with elevated radii. Did the authors want to say they were more widespread at a larger radius? In addition, the authors should provide close-up figures showing the details of difference between “crack” and “shear damage” areas.

15.  Page 8, Line 239 – 240

The authors wrote, “These behaviors could be attributed to the brittle nature of sample, indicating imperfect interphase bonding between fiber-epoxy interactions”.

What does it mean by saying “imperfect interphase bonding”? Did the authors want to say curing/crosslinking on the interfaces? While vacuum resin infusion is primarily a mechanical and manufacturing process rather than a chemical process, the authors should be careful about phrasing. Bonding is typically used for a chemical reaction, so it would be better to say weak interface interaction than interfacial bonding.

16.  Page 9, Line 245 – 246

From Figure 10, can the authors explain why there are oscillations in the torque-angle curves for the FEM models after failure? Again, what are the strain rates and torsion rates in the FEM simulations? It seems the authors applied ultra-high rates, which caused the shock wave propagation through the FEM models. Please note that if shock wave propagation were to occur in a torsion test, it would not accurately reflect the behavior of the material under the experimentally accessible loading conditions.

17.  Page 9 - 10, Line 266 – 274

Can the authors provide the legend for colored regions that appear in the notched samples? What red, blue, and cyan regions stand for?

18.  Page 10, Line 278 – 279

Again, the authors need to explain each symbol and parameter in the formula.

19.  Page 10, Line 285 – 298

(1)  The subsection titled “The torsion mechanism” does not provide a satisfactory analysis or discussion of the torsion mechanisms. The relevant theoretical references could be added to support the analysis. Additionally, the mentioned paragraph fails to offer any suggestions for potential solutions or improvements to address the observed weaknesses in the interlaminar bonding strength and brittleness of the samples.

(2)  The colors of the highlighted boxes in Figure 13 do not align with the legend for delamination. Can the authors modify either the legend or the markers to resolve this discrepancy?

(3)  Did the authors want to say, “"first principal stress σ1”?"? “First Principle(s)” has the other meaning in physics.

(4)  Did the authors want to say, “permanent unrecoverable plastic deformation” other than “permanent uncoverable plastic deformation”? Again, please carefully check out the wording and rephrasing before the next submission.

(5)  It is unclear what the numbers on the paper tags indicate. Furthermore, the handwriting on the tag labeled “V-30°” is illegible due to damage. Can the authors offer an explanation or provide a replacement tag?

20.  Textual issues and grammatical errors

(1)  Page 1, Line 16

The authors wrote, “Variation of notched properties were explained…”. Please correct “were” to “was”.

(2)  Page 1, Line 34

The authors wrote, “A great amount of experiments have been performed…”. Please correct “a great amount of” to “a great number of”, since “experiment” is a countable noun.

(3)  Page 2, Line 58

Please correct “et al” to “et al.”.

(4)  Page 3, Line 103

The “2” in the unit “g/cm2” should be superscripted.

(5)  Page 3, Line 119

The authors wrote “tension tests”, but in the previous text the authors wrote “tensile tests”. Please be consistent with the use of such terms.

(6)  Page 7, Line 218 – 219

The authors wrote, “…, which is contribute to an enhancement in their tensile strength of C-notched samples”. Please rephrase this sentence since it does not read grammatically correct.

(7)  Page 7, Line 222 – 223

The authors wrote, “…, which is contribute to an enhancement in their tensile strength of U-type samples”. Please rephrase this sentence since it does not read grammatically correct.

(8)  Page 10, Line 287 – 288

The authors wrote, “…, which is induced by the first principle stress σ1 reaches up to the critical strength of fibers under the torsion loading.” Please rephrase this sentence since it does not read grammatically correct.

(9)  Tense errors

There are multiple instances where the authors misused or mixed up tenses between the present and the past, e.g., Page 6, Line 206 – 208, the authors used present tense in the first half of the sentence but the past tense in the second half. I suggest that the authors should carefully run proofreading and grammar check throughout the manuscript before the next submission.

The formatted comments can be also found in the PDF file. Thanks!

Author Response

  • Page 1, Line 9 – 18: The Abstract does not provide sufficient details on the specific findings or implications of the study. I suggest that the authors should add a sentence or two to discuss the main results (with specific values reported) as well as the broader impact of this research to make the Abstract more informative.

Answer:

The tensile strengths of C-notch and U-notch increase with increasing notch radius, while the ultimate torque of C-notch and V-notch decrease with increasing notch size and angle.

  • Page 1 – 2, Line 34 – 72,
    (1) The authors presented several experimental studies examining the strength reduction of notched CFRP laminates. However, the authors did not report specific numerical values regarding the extent to which the strength was reduced.
    (2) The authors are expected to limit their literature review on the residual strength of notched CFRP laminates with respect to shape and size under different loadings to the most recent five years, in order to ensure that the references accurately reflect the current state of knowledge in the field.

Answer:

The specific datum in these relevant references are added in the revised version.

  • Page 3, Line 110 – 120,
    In the subsection titled “Manufacturing of composites”, the authors did cover the manufacturing process and testing method for producing CFRP laminates, but additional details regarding the materials, equipment, and specific testing procedures are expected to be included to enhance the understanding of such a process. For instance, elaborating on the type of equipment used for vacuum resin infusion process.

Answer:

Some details in making the samples were added.

  • Page 4, Line 124 – 127: Figure 2 and Figure 3 appear to be duplicated, since the shape of each notched specimen can be inferred from Figure 2. Maybe the authors can only keep Figure 2 or merge these two figures into one.

Answer:

Corrected

  • Page 4, Line 129 – 135: Can the authors elaborate on the meanings of ASTM and D638 standards?

Answer:

ASTM D638 for testing tensile properties of plastics

ASTM D3039 used for testing tensile properties of polymer matrix composite materials.

  • Page 4, Line 142 – 144,
    (1) Figure 4 should include labels for subfigures. For example, subfigure labels (a), (b), (c), and (d) could be added to show different views of the specimen. Axes indicating longitudinal and transverse directions could also be added.
    (2) Readers are unable to readily discern the type of test being conducted or distinguish between the top right and bottom right schematics. To address this, the authors may consider adding subfigure labels (e) and (f) (and the associated descriptions in the caption), as well as double-ended arrows indicating the direction of torsion in these subfigures.

Answer:

(1) Corrected

(2) The necessary explanations were added in the revised version.

  • Page 5, Line 145 – 167
    (1) Can the authors provide visualizations of the mesh they used in their FEM models? Specifically, I am interested in how the authors treated discontinuities in the notched areas and cracks.
    (2) What are the strain rates and the torsional angle rates used in the FEM simulations? Are these parameters and setups comparable with those in the actual experiments?
    (3) Can the authors provide an explanation for why delamination failure was not taken into account?

Answer:

Mesh convergence is tested to ensure computational accuracy.

Because the mechanical performance is considered to be static analysis, the strain rate and the torsional rate were not assigned in FEM modeling.

The delamination damage should be important in the torsion deformation, and affects the torsion failure in the present work, but the modeling is very difficult to account for the ply delamination.

  • Page 5, Line 170 – 174: The authors are expected to provide detailed explanations for symbols and parameters that appear in equation (1) through (5).

Answer:

To simulate the above failure modes, the elastic properties are made to be dependent on three field variables, FV1, FV2, FV3 and FV4. Four variables represent matrix failure, fiber/matrix shearing failure, fiber failure and damage, respectively. The values of the field variables are set equal to zero in the undamaged state. After a failure index has exceeded 1.0, the associated user-defined field variable is set equal to1.The mechanical properties in the damaged area are reduced appropriately, according to the property degradation model defined in Table 2. For example, if the matrix failure criterion satisfied, namely σ 2 ≥ 0, e 2 m ≥ 1, then only degrade E22→0, ν12 →0, while E11 , G12 , G13 , G23 equals to the initial value.

Table 2. The stiffness degradation rules based on the failure state (1-fiber direction, 2-transverse direction, FV1-matrix crack, FV2-fiber/matrix shear failure, FV3-fiber collapse, FV4-damage)

Initial

Matrix crack

Fiber/matrix shear

Fiber break

Damage

E11

E11

E11

E110

E11

E22

E220

E22

E220

E22

ν12

ν120

ν120

ν120

ν12

G12

G12

G120

G120

G120

G13

G13

G130

G130

G13

G23

G23

G23

G230

G23

FV1=0

FV1=1

FV1=0

FV1=0

FV1=0

FV2=0

FV2=0

FV2=1

FV2=0

FV2=0

FV3=0

FV3=0

FV3=0

FV3=1

FV3=0

FV4=0

FV4=0

FV4=0

FV4=0

FV4=1

  • Page 6, Line 181 – 187
    (1) The authors wrote, “Comparing the predictions with the measured results could determine the basic material properties, …”. It is not clear how the authors compared the predictions with the measured results.
    (2) Are the material properties provided in Line 184 – 185 obtained through experimentation? If yes, then how did the authors calculate the Young's modulus given that the stress-strain curves appear to have a non-linear elastic region? Did the authors perform linear regression or take the second derivatives of the polynomial fitting to calculate it?
    (3) There is no explanation to tell the difference between Exp. 1 and Exp. 2 in Figure 5.
    (4) Can the authors provide an explanation as to why the experimental curves cease to exhibit further data prior to the point of failure in Figure 5?

Answer:

1) The used material properties in the present simulations are quoted from the other work [34]

2) This problem is similar to the above

3) (exp.1 and exp.2 denote two samples with same sizes)

4) The stiffness degradation method allows us to obtain the stress softening stage after the failure point in Figure 4.

  • Page 6, Line 188 – 198
    (1) The authors stated, “…, tensile strength of C25 sample exhibited the highest value that is nearly similar to that of unnotched samples.” Is this conclusion drawn from experiments or from FEM simulations? Since the experimental ultimate strength values were not presented or discussed.

Answer:

Both numerical simulations and testing results present that all the circular notched samples exhibit a similar stress-strain relations, and C25 sample possesses the highest tensile strength that is nearly similar to that of unnotched samples.

  • The authors should report all these kinds of specific values in the discussions throughout Section 4.
    (2) The authors stated, “…, while the numerical calculations overestimate the tensile stiffness for the V-notched laminates, and present lower predictions to their failure strengths.” The authors should conduct a detailed examination and discussion on the possible factors that contribute to the overestimation of stiffness. Moreover, the authors should provide more information on the potential errors or uncertainties from both the experiments and the simulations.

Answer:

Furthermore, the stress-strain relations in the beginning stage of stretching are nonlinear, accompanying the marked transition from high stiffness to low stiffness. Such a change maybe originate from the inaccuracy in ply orientation, off-centering clamp and sample cutting. The prediction errors in the ultimate strength between numerical and measured results are in range of 1~5% for C-notch, and ranges from 2% to 29.4% for U-notch, while almost 14% underestimation for V-notch.

  • Page 6, Line 199 – 200: The labels are included in subfigures of Figure 6, but the letters do not appear in the caption.

Answer:

Added

  • Page 6, Line 204 – 205
    The authors mentioned that “the changing tendency of strength with the notch radius is in accordance with that is found in Morgan’s work”. Can the authors provide a clear explanation of the changing tendency in this context? Additionally, can the authors elaborate on the findings from Morgan's work? The current discussion section reads overly general and ambiguous, so a more informative and specific discussion would be expected.

Answer:

In Morgan’s researches, the root radius of U-notched specimens changes from 0.5mm to 20mm, and the corresponding failure force increases from 4.19 kN up to 4.82 kN. Therefore, their dependence of tensile strength on the root radius is the same as our results.

Moreover, it should be mentioned that the changing tendency in the failure torque of U-notched samples with the notch radius is also in agreement with that in Morgan’s work [4]. In Morgan’s researches, the root radius of U-notched specimens changes from 0.5mm to 20mm, and the corresponding failure torque increases from 2.76 N·m up to 3.66 N·m. Therefore, their dependence of torsion strength on the root radius is the same as our results.

  • Page 7, Line 211 – 212
    Page 9, Line 257 – 258
    (1) Can the authors modify the x-axis ticks for Figure 7 (b) and Figure 11 (b)? The authors omitted 3 and 5, so readers cannot identify U-3 and U-5 from the horizontal ticks in Figure 7 (b) and Figure 11 (b).
    (2) Again, the authors should include the letter labels in the captions of Figure 7 and 11.
    (3) The authors presented the bar plots with errors in Figure 7 and Figure 11, but there is no description provided regarding the methods/steps to calculate these errors, e.g., the number of samples considered in the statistical calculations.

Answer:

1) This suggestion is very good for this paper.

2) Added.

3) The explanations are added: here the errors are determined by measuring at least three specimens with the same sized notches.

  • Page 7, Line 213 – 219
    (1) Figure 8 lacks a color code or legend to indicate the corresponding representation of the blue and red regions.
    (2) The authors stated, “Figure 8 shows the predicted damage modes in C-notched specimens”. However, Figure 8 also shows the predicted damage modes in U-notched and V-notched laminates. In addition, the authors should discuss how the positions and distributions of the damage modes were predicted.
    (3) The authors stated, “both fiber crack and shear damage areas gradually reduced with increasing the notch radius from 10 mm to 25 mm”. It is hard to see the fiber crack and shear damage areas gradually reducing with elevated radii. Did the authors want to say they were more widespread at a larger radius? In addition, the authors should provide close-up figures showing the details of difference between “crack” and “shear damage” areas.

Answer:

1) Added a legend.

2) Modified according to the comments.

3) Become more and more wide along the C-shape notch with increasing the notch radius from 10 mm to 25 mm, and the severity of damage is relatively reduced, and therefore their corresponding tensile strengths are increasing with the decrease in these damage modes.

  • Page 8, Line 239 – 240
    The authors wrote, “These behaviors could be attributed to the brittle nature of sample, indicating imperfect interphase bonding between fiber-epoxy interactions”. What does it mean by saying “imperfect interphase bonding”? Did the authors want to say curing/crosslinking on the interfaces? While vacuum resin infusion is primarily a mechanical and manufacturing process rather than a chemical process, the authors should be careful about phrasing. Bonding is typically used for a chemical reaction, so it would be better to say weak interface interaction than interfacial bonding.

Answer:

Many thanks for the reviewer’s suggestion, and we learned a lot from this comment.

These behaviors could be attributed to the brittle nature of epoxy resin, and imperfect interphase cohesion between carbon fiber and epoxy matrix. Failure mechanisms over the fracture surface indicate that some carbon fibers pull out, exhibiting the weak interfacial interaction induced by the vacuum infusion, and some micro-voids are still in existence. Moreover, the adhesion between fiber and epoxy mainly stems from the mechanical friction between them without any chemical bonding.

  • Page 9, Line 245 – 246
    From Figure 10, can the authors explain why there are oscillations in the torque-angle curves for the FEM models after failure? Again, what are the strain rates and torsion rates in the FEM simulations? It seems the authors applied ultra-high rates, which caused the shock wave propagation through the FEM models. Please note that if shock wave propagation were to occur in a torsion test, it would not accurately reflect the behavior of the material under the experimentally accessible loading conditions.

Answer:

This comment is very important to this work and very helpful to deep understanding.

These predicted torque-angle curves present a marked oscillations in the torque at the failure stage, which lies in the damage accumulations during the torsion deformation.

  • Page 9 - 10, Line 266 – 274
    Can the authors provide the legend for colored regions that appear in the notched samples? What red, blue, and cyan regions stand for?

Answer:

Red delegates SDV=1, and blue delegates SDV=0 in these contours

  • Page 10, Line 278 – 279
    Again, the authors need to explain each symbol and parameter in the formula.

Answer:

G is shear modulus, and is polar moment of inertia over the minimum transaction of the laminates, in which ρ is the distance from the shaft center to element of infinitesimal area dA.

  • Page 10, Line 285 – 298 (1) The subsection titled “The torsion mechanism” does not provide a satisfactory analysis or discussion of the torsion mechanisms. The relevant theoretical references could be added to support the analysis. Additionally, the mentioned paragraph fails to offer any suggestions for potential solutions or improvements to address the observed weaknesses in the interlaminar bonding strength and brittleness of the samples.
    (2) The colors of the highlighted boxes in Figure 13 do not align with the legend for delamination. Can the authors modify either the legend or the markers to resolve this discrepancy?
    (3) Did the authors want to say, “"first principal stress σ1”?"? “First Principle(s)” has the other meaning in physics.
    (4) Did the authors want to say, “permanent unrecoverable plastic deformation” other than “permanent uncoverable plastic deformation”? Again, please carefully check out the wording and rephrasing before the next submission.
    (5) It is unclear what the numbers on the paper tags indicate. Furthermore, the handwriting on the tag labeled “V-30” is illegible due to damage. Can the authors offer an explanation or provide a replacement tag?

Answer:

1) This comment is very important, but it is very difficult to answer this question at the presnet time for us.
Torsion provides a more complete picture of composite deformation for specific service scenarios as a complex deformation mode. But the analysis of multi-damage progression under torsion should be challenging for CFRP laminates, which demands a comprehensive understanding in the next work.

2) Corrected.

3) Yes, this is first principal stress σ1.

4) Corrections were conducted.

5) In our torsion test, we found that most of V-notched specimens could experience a torsion angle larger than 300 degree before the final failure of the whole CFRP laminate, and also our numerical results exhibit a distortion in shape.

  • Textual issues and grammatical errors
    (1) Page 1, Line 16 The authors wrote, “Variation of notched properties were explained…”. Please correct “were” to “was”.
    (2) Page 1, Line 34 The authors wrote, “A great amount of experiments have been performed…”. Please correct “a great amount of” to “a great number of”, since “experiment” is a countable noun.
    (3) Page 2, Line 58 Please correct “et al” to “et al.”.
    (4) Page 3, Line 103 The “2” in the unit “g/cm 2 ” should be superscripted.
    (5) Page 3, Line 119 The authors wrote “tension tests”, but in the previous text the authors wrote “tensile tests”. Please be consistent with the use of such terms.
    (6) Page 7, Line 218 – 219 The authors wrote, “…, which is contribute to an enhancement in their tensile strength of C-notched samples”. Please rephrase this sentence since it does not read grammatically correct.
    (7) Page 7, Line 222 – 223 The authors wrote, “…, which is contribute to an enhancement in their tensile strength of U-type samples”. Please rephrase this sentence since it does not read grammatically correct.
    (8) Page 10, Line 287 – 288 The authors wrote, “…, which is induced by the first principle stress σ1 reaches up to the critical strength of fibers under the torsion loading.” Please rephrase this sentence since it does not read grammatically correct.
    (9) Tense errors: There are multiple instances where the authors misused or mixed up tenses between the present and the past, e.g., Page 6, Line 206 – 208, the authors used present tense in the first half of the sentence but the past tense in the second half. I suggest that the authors should carefully run proofreading and grammar check throughout the manuscript before the next submission.

Answer:

Many thanks for the reviewer’s effort in checking these errors in our work.

Corrected

Reviewer 4 Report

This paper presents the notched behaviours of carbon fibre reinforced epoxy matrix composite laminates. the The following points need to be clarified for paper to be considered for the publication:

  1. The abstract needs to be clarified for “residue strength (tension and torsion)”. Are they residual strength?
  2. Why did the authors select torsional experiments? Its importance needs to be given in the section “Introduction”.
  3. The following papers could be accommodated for the completeness and update of the literature review

https://doi.org/10.1016/j.engstruct.2022.115250

https://doi.org/10.1016/j.engfracmech.2021.107802

4.      It is suggested that the authors should modify the last part of the introduction to: 1) clearly mention the goal and novelty of this work, 2) mention the methodology used and its important to place the major hypothesis, 3) mention the structure of the paper or procedure of their work and expected results.

5.      No information about the property degradation model was given. Was damage evolution used in the model? If so, what are the fracture energy values?

6.      No information about the FE models of the tests are given? Mesh size, its convergency, BCs, etc.

  1. How did the authors find so many parameters in Section 4.1. with one experimental curve? Normally you should take them from the literature and demonstrate that it fits well to experiments. The authors should comment on this.
  2. How about the deviations of the predictions from the experiments in Fig. 6, especially in Fig. 6(c)?

9.      The paper lacks a discussion on the weakness and limitations of the present methodology/study. Comparison of Experiments and Simulation in Fig. 10 needs to be reconsidered in connection with the appropriateness of Hashin Model. How about other models? LaRC05, Puck, etc.

Author Response

  • The abstract needs to be clarified for “residue strength (tension and torsion)”. Are they residual strength?

Answer:

The residue strength in the original version was modified as the failure strength in the revised version to avoid the misreading.

  • Why did the authors select torsional experiments? Its importance needs to be given in the section “Introduction”.

Answer:

The torsion testing on the notched CFRP seems very few at present time, and insufficient for structural engineering.

  • The following papers could be accommodated for the completeness and update of the literature review

Answer:

Corrected

  • It is suggested that the authors should modify the last part of the introduction to: 1) clearly mention the goal and novelty of this work, 2) mention the methodology used and its important to place the major hypothesis, 3) mention the structure of the paper or procedure of their work and expected results.

Answer:

1) the purpose of this work is to reveal the effect of notch size and shape on the mechanical behaviors of CFRP laminates under tension and torsion, especially the torsion testing on the notched CFRP seems very few at present time, and insufficient for structural engineering. 2) A joint method of experiment and FEM was used to understand the damage mechanism in these notched samples. 3) In Section 2, we present the preparation of samples for tensile and torsion testing. The user subroutine and FEM modeling is then presented in Section 3. The results of both measured and simulated, failure mechanisms under tension are discussed in Section 4. The results of both tested and simulated, failure mechanisms under torsion are discussed in Section 5. The conclusion is finally given in Section 5.

  • No information about the property degradation model was given. Was damage evolution used in the model? If so, what are the fracture energy values?

Answer:

This comment is same as Q11, and the corresponding modification was given.

  • No information about the FE models of the tests are given? Mesh size, its convergency, BCs, etc.

Answer:

This comment is similar to Q10, and some details were added in the revised paper.

  • How did the authors find so many parameters in Section 4.1. with one experimental curve? Normally you should take them from the literature and demonstrate that it fits well to experiments. The authors should comment on this.

Answer:

The reference was added

[34] Duan, M.M.; Shi W.Y.; Zhang, X.Y. Numerical Analysis and Tests of Composite Laminates under Low-velocity Impact. Structure & Environment Engineering 2020, 26-31.

  • How about the deviations of the predictions from the experiments in Fig. 6, especially in Fig. 6(c)?

Answer:

Based on the comparisons between numerical results and testing datum, the present simulations could predict the tested failure strengths for C-notch and U-notch samples, while the adopted numerical method for V-notch specimens under-estimate the measured results.

  • The paper lacks a discussion on the weakness and limitations of the present methodology/study. Comparison of Experiments and Simulation in Fig. 10 needs to be reconsidered in connection with the appropriateness of Hashin Model. How about other models? LaRC05, Puck, etc.

Answer:

Based on Hashin’s damage criterion, the in-plane multi-damage modes in CFRP laminates can be well assessed by directly programing a user-subroutine in the ABAQUS code. However, the out of plane damage mode cannot be considered by this model, and thus an additional cohesive zone with traction-separation separation laws should be involved to consider delamination growth, especially for the complicated multiply failure progress in the torsion loading.

Round 2

Reviewer 1 Report

The revised manuscript answered a lot of questions on the previous version, but still remains some unanswered queries and some more comments occur. They are as follows.

1.      Please double check Eq. (2), in which no fiber-axial stress component occurred.

2.      On page 6, in the line 206, “…on three field variables, FV1, FV2, FV3 and FV4”, why was “three” but not “four”?

3.      What did “Damage” in the last column of Table 2 stand for? In the manuscript, “damage” and “failure” were not rigorously defined. All the four field variables, FV1, FV2, FV3 and FV4, were set to zero if no damage occurred. Thus, a “damage” can refer to any kind of failure. The authors need to strictly define the damage in the last column of Table 2. Similarly, the statement in the line 207 of page 6, “…, fiber failure and damage, …”, needs to be defined clearly.

4.      It is assumed that an incremental step-by-step loading process was applied by the authors in their simulation. What was the termination condition for the authors to obtain the simulation curve plotted in Fig. 4?

5.      What was the termination condition used in simulating the curves in Fig. 5 and Fig. 9?     

Author Response

Correspondence to the reviewers’ comments

  • Please double check Eq. (2), in which no fiber-axial stress component occurred.

Answer:

This is a very important suggestion.

We corrected this error is Equation (2).

  • On page 6, in the line 206, “…on three field variables, FV1, FV2, FV3 and FV4”, why was “three” but not “four”?

Answer:

We carefully checked this section and modified these mistakes in the original version.

  • What did “Damage” in the last column of Table 2 stand for? In the manuscript, “damage” and “failure” were not rigorously defined. All the four field variables, FV1, FV2, FV3 and FV4, were set to zero if no damage occurred. Thus, a “damage” can refer to any kind of failure. The authors need to strictly define the damage in the last column of Table 2. Similarly, the statement in the line 207 of page 6, “…, fiber failure and damage, …”, needs to be defined clearly.

Answer:

Many thanks for the reviewer, and this is an important comment.

We carefully sorted out this part, and corrected some serious errors in the original text.

  • It is assumed that an incremental step-by-step loading process was applied by the authors in their simulation. What was the termination condition for the authors to obtain the simulation curve plotted in Fig. 4?

Answer:

There is no any termination condition in the present numerical method. The damage progression was described by using continuous damage mechanics, and therefore the stress-strain responses are developing continuously.

  • What was the termination condition used in simulating the curves in Fig. 5 and Fig. 9? 

Answer:

There is no any termination condition in the present numerical method. The damage progression was described by using continuous damage mechanics, and therefore the stress-strain responses are developing continuously. 

Reviewer 2 Report

Dear Authors, this reviewer raised an issue that the present study doesn't matched between simulation and experimental result i.e. tensile strength. This reviewer couldn't find the response on this, so nothing to review again.

Author Response

  • Dear Authors, this reviewer raised an issue that the present study doesn't matched between simulation and experimental result i.e. tensile strength. This reviewer couldn't find the response on this, so nothing to review again.

Answer:

Thank you for this piece of comment.

In the revised version, we added an amount of analyses on the comparisons between the predictions and tested results, and their differences were also explained from the existing limitations in the present numerical modelling.

Reviewer 3 Report

The authors have effectively addressed my questions and comments in the revised version, although there are a few challenging issues remaining unresolved in the current stage. One significant issue in the response letter is that the authors need to specify the locations in the revised manuscript where they have made the corrections or additions, instead of saying "corrected" or so. I recommend that the authors consider doing so in future publications.

Author Response

  • The authors have effectively addressed my questions and comments in the revised version, although there are a few challenging issues remaining unresolved in the current stage. One significant issue in the response letter is that the authors need to specify the locations in the revised manuscript where they have made the corrections or additions, instead of saying "corrected" or so. I recommend that the authors consider doing so in future publications.

Answer:

Many thanks for the reviewer’s suggestions, which are good food for us.

Reviewer 4 Report

The points are revised reasonably.

Author Response

  • The points are revised reasonably.

Answer:

The authors want to express sincere gratitude to the reviewer’s comments.

Round 3

Reviewer 1 Report

All of my other queries have been answered, except for the termination conditions. As long as a stress-strain curve was predicted, a termination or calculation-stop condition must have been employed by the authors. Otherwise, the curve plot would be lasted forever. It seemed that the authors predicted the curve bigger than the measured counterpart and only a partial of the predicted curve was plotted in the figure. If so, the authors need to add a description in the paper. 

Author Response

It should be noted that a termination condition is not involved in the present modeling method, only the failure criterions are incorporated to monitor these failure modes. After the damage happens, the iteration increment will be automatically selected and decreased gradually with increasing the damage degree. As the iteration increment is less than the predetermined minimum increment size, the computation process will be terminated correspondingly.

Reviewer 2 Report

This reviewer suggest accept as is.

Author Response

many thanks for the reviewer's comments!

Round 4

Reviewer 1 Report

Please would the editor change the statement in the lines 214 through 219, i.e. "It should be noted that a termination condition is not involved in the present modeling method, only the failure criterions are incorporated to monitor these failure modes. After the damage happens, the iteration increment will be automatically selected and decreased gradually with increasing the damage degree. As the iteration increment is less than the predetermined minimum increment size, the computation process will be terminated correspondingly.", into the following one:

It is noted that after a damage happens, the iteration increment will be automatically selected and decreased gradually with increasing the damage degree. As the iteration increment is less than the predetermined minimum increment size, the computation process will be terminated correspondingly.

If necessary, the editor may send the change to the authors for approval. No need to send the manuscript to me for a further going through. 

Author Response

we sincerely adopted the reviewer's suggestion.

here all the authors want to express our sincere gratitude to you for your rigorous academic attitude.